# An Analysis of Waste Heat Recovery from Wastewater on Livestock and Agriculture Farms

**Daniel Słyś \***, **Kamil Pochwat** **and Dorian Czarniecki**

Department of Infrastructure and Water Management, Rzeszow University of Technology, al. Powstańców Warszawy 6, 35-959 Rzeszów, Poland; kp@prz.edu.pl (K.P.); Dorian.Czarniecki@interia.pl (D.C.)
**\*** Correspondence: daniels@prz.edu.pl; Tel.: +48-17-865-1748

**Abstract:** Agriculture is one of the sectors of the economy in which it is possible to conduct much more rational energy economy. The easiest way to achieve financial savings as well as reduce air pollution is to use waste heat sources. Heat pumps are perfect for this. Particularly favorable is the case when the device can operate in an alternative system and serve both heating and cooling purposes. The purpose of this article was to present possible solutions for installations enabling heat recovery from wastewater to supply agri-breeding farms with hot utility and technological water, a financial analysis of their application, and an assessment of the impact of these solutions on possible reduction of pollutant emissions. The tests were carried out for four variants of cooperation between a heat pump and an exchanger. In the first variant, waste heat was used in the process of heating water used to clean stands and prepare feed. In the second variant, waste heat took part in heating the water used for watering plants. In the third variant, waste heat was used in the process of drying cereals. In turn, in the last variant, waste heat supported the preparation of utility hot water for the breeder's residential building. The study showed the legitimacy of using thermal energy from liquid manure as a waste heat source on farms and farming. This is mainly due to the short payback period, which can be within 2–4 years. In turn, the analysis of pollution reduction associated with the recovery of waste energy showed that the use of heat pumps allowed a significant reduction in the emission of harmful compounds to the atmosphere, in particular carbon dioxide. It is worth noting that livestock breeding is one of the most important branches of agricultural production not only in Poland but also throughout Europe, Asia and South and North America. For this reason, the use of waste heat-recovery systems enables real savings in the purchase of energy and reduction of pollutant emissions arising during traditional production processes.

**Keywords:** waste heat recovery; heat pump; agriculture; financial analysis; air pollution

## 1. Introduction

Since the beginning of the twentieth century, a clear development of the population can be observed which results in a huge demand for food, including products of animal origin. Consumer life led by residents of developed and developing countries make agriculture and animal husbandry one of the most important sectors of the economy.

Modern agriculture faces many difficult challenges. On the one hand, there is an urgent need to develop modern technological solutions that would guarantee continuous cheap production of food with high-quality parameters [1], on the other one, there is a need to reduce the negative impact on the natural environment [2]. Currently, there is no doubt that agriculture, like any production activity, is a real threat to the environment [3,4], and reducing this threat is one of the priorities of the modern food production economy.

The obligation to improve energy efficiency and increase the use of renewable energy sources results from the energy and climate policies adopted in many countries. The current direction of development in terms of the energetics of the European Union countries results from the adopted law policy. The most important legal regulations are: Directive 2009/28/EC on the promotion of the use of energy from renewable sources [5] and Directive 2009/29/EC amending Directive 2003/87/EC in order to improve and extend the Community scheme for greenhouse gas emission allowance trading [6], included in the climate and energy package. In turn, the need for energy-efficiency measures is regulated by Directive 2012/27/EU on energy efficiency [7]. This applies to the activities of the state, local governments, enterprises, households and agri-breeding farms.

The idea of sustainable development emphasizes the search for new solutions in heat supply systems, such as the development of low-emission technologies of energy production, especially from waste sources and an improvement of energy production efficiency [8,9]. The answer to this trend on farms and farming is the use of low-temperature waste energy from liquid manure using heat pumps.

Rising prices of fuel and awareness of responsibility for the natural environment are conducive to growing interest in the use of waste energy sources [10–12]. A high-efficiency heat pump can be used to recover this energy. Slurry is a waste product with a liquid consistency formed during animal husbandry without using litter, being a mixture of solid and liquid animal faeces with the addition of process water used for flushing [13,14]. Due to the rich composition, it is a valuable natural fertilizer [15,16].

Slurry can also be an extremely desirable bottom energy source for heat pumps with its retention, e.g., in concrete underground tanks [17]. It is always at a temperature higher than the ambient temperature. Thanks to this it is possible to achieve high seasonal efficiency. Heat recovery from liquid manure by a heat pump, causing it to cool, brings a number of measurable benefits for financial and environmental reasons and does not adversely affect animal health. An important environmental aspect is worth noticing. Lowering slurry temperature slows down putrefaction and reduces the amount of gases formed, including greenhouse gases, such as methane, carbon dioxide, hydrogen sulfide and ammonia [18,19]. Not without significance is the fact that the operating heat recovery installation does not emit any pollution, does not require supervision, and is characterized by a stable operation throughout the year, which affects the high financial effect of the system and production farm.

Agricultural production requires the supply of large amounts of low-temperature heat consumed, among others, in:

- heating utility water for technological purposes in food production technologies and for sanitary purposes in households;
- heating utility water for irrigation of plants produced under cover;
- heating utility water in livestock buildings for watering and preparing animal feed;
- heating air in drying equipment and vegetable and fruit storage rooms.

Farm heat recovery is the subject of research undertaken by interdisciplinary research teams around the world. Most of the current research conducted is carried out in the environmental, economic and production aspect (Figure 1).

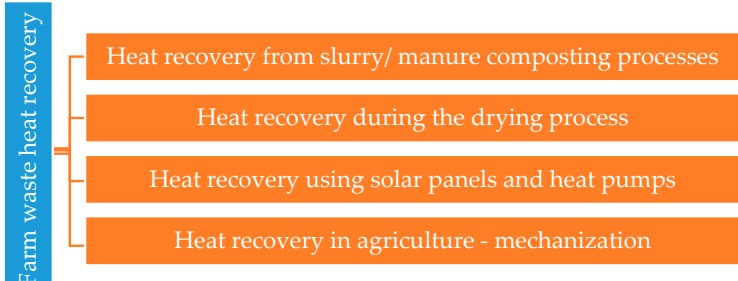

**Figure 1.** Waste heat-recovery area.

The goals and directions of the research conducted in recent years related to the use of waste heat in agriculture are summarized in Table 1.

**Table 1.** Farm waste heat recovery.

| Heat Recovery Area | Aim of Research | Author |
|---|---|---|
| **Heat recovery from slurry/manure composting processes** | The authors of the paper focused on analyzing the possibilities of waste heat recovery from discharged slurry which can be used to improve the efficiency of biogas production. In the research, the authors focus on energy recovery from waste of animal/agricultural origin. The research focused on replacing conventional energy sources to reduce the use of non-renewable raw materials. In their research, the authors analyzed the efficiency of the conversion system that allows obtaining waste heat energy from animal excrements from combined drying and combustion processes. | Chen, J.; Ma, C.; Ji, X.; Lu, X.; Wang, C. [20] Gheorghe, L.; Pană, C.; Mihaescu, L.; Cernat, A.; Negurescu, N.; Mocanu, R.; Negreanu, G. [21] Have, H.; Fritze, M. [22] |
| **Heat recovery during the drying process** | An analysis of heat recovery efficiency in drying processes. In the research the authors analyzed the amount of waste heat recovered from the ginger drying process in a biomass convection dryer with heat storage material (SHSM) and phase change material (PCM) | Pati, J.; Hotta, S. [23] |
| **Heat recovery using solar panels and heat pumps** | The research showed significant financial savings related to the use of renewable energy through solar panels. The authors of the study attempted to assess the selection of the heat source for the greenhouse facility on farms. Their research showed that the use of heat pumps could be considered a competitive method of heating greenhouse facilities due to greater economic efficiency and in relation to environmental protection. In the research, the authors made a 3D model of a ground heat exchanger. The simulations carried out confirmed the legitimacy of using this type of facilities in agriculture. | Liu, Y.M.; Chang, K.C.; Lin, W.M.; Chung, K.M. [24] Nemś, A.; Nemś, M.; Świder, K. [25] Deglin, D.; Caenegem, L.; Dehon, P. [26] |
| **Heat recovery in agriculture— mechanization** | Development of a heat recovery model from Euro VI class internal combustion engines, enabling an increase in waste energy recovery efficiency by 15%. The article examined the technical and economic possibilities of heat recovery from agricultural machinery. The article demonstrated the economic legitimacy of using waste heat in agricultural machinery, especially in winter periods to reduce fuel consumption. An analysis of the use of the Organic Rankine Cycle (ORC)—a technology of converting medium and low-quality waste heat into mechanical energy and electricity in natural gas engines. Actions taken were aimed at reducing fuel consumption, thus reducing the operating costs of mechanical devices in agriculture and environmental protection by improving air quality. | Feru, E.; Willems, F.; Jager, B.; Steinbuch, M. [27] Kalinichenko, A.; Havrysh, V.; Hruban, V. [28] Valencia, G.; Fontalvo, A.; Cardenas E.,Y.; Duarte, J.; Isaza-Roldan, C. [29] |

Recovery and use of waste heat in agriculture has great development prospects and significant quantitative potential, which, however, has not yet been sufficiently utilized. Therefore, there is a need for further research aimed at maximizing the positive effects associated with the use of waste heat in agriculture, including the effects of reducing greenhouse gas emissions. Taking this into consideration, this article analyzes various technical solutions enabling the utilization of waste heat from liquid manure. The research conducted proved that the way waste energy is used significantly affects the economic and environmental effects obtained. It is worth paying attention to the need to perform technical and financial analyzes when choosing specific technical solutions. Solutions that seem to be the most beneficial at first, in many cases, after a deeper analysis, may prove financially ineffective.

## 2. Test Object and Input Data

Energy and financial efficiency tests were carried out for a complex of agricultural production facilities. The data adopted for the simulation are presented in Table 2.

**Table 2.** Input data for analysis.

| Input Data | Type of Data | Value |
|---|---|---|
| **Construction data** | a livestock building area | 600 m$^2$ |
| | a livestock building high | 6 m |
| | a monolithic underground slurry tank capacity | 450 m$^3$ |
| | a dryer area | 51 m$^2$ |
| | a greenhouse area | 1000 m$^2$ |
| **Design parameters of the system cooperating with the heat pump** | number of inhabitants in the household, $M$ | 6 |
| | number of piglet breeding places | 400 |
| | demand for water for cleaning stands, watering and preparing feed for pigs, $q_{śr\ d1}$ | 4000 L/day |
| | water demand for plant production, $q_{śr\ d2}$ | 5000 L/day |
| | water demand for a residential building, $q_{śr\ d3}$ | 660 L/day |
| | system operation time per day | 18 h |
| | coefficient of hourly irregular distribution of water | 2.5 |
| | municipal water temperature, $T_z$ | 10 °C |
| | water temperature for cleaning stands, watering and preparing animal feed, $T_p$ (variant I), | 60 °C |
| | irrigation water temperature, $T_n$ (variant II), | 20 °C |
| | water supply temperature to the water air heater, $T_z$ (variant III), | 55 °C |
| | DHW temperature, $T_c$ (variant IV), | 50 °C |
| | lower heat source—sewage at a temperature of $T_{śc}$ (average sewage temperature) [19,23], | 14 °C |
| | specific heat of water, $c_w$ | 4174 J/kg·K |
| **Input data for economic analysis** | system life, which corresponds to the time of trouble-free operation of heat pump compressors, was assumed at, $N$ | 20 years |
| | discount rate, $p$ | 6% |
| | electricity price, $c_{el}$ | 0.14 €/kWh |

## 3. Variants

As part of the research, energy and financial analysis was performed for four variants of cooperation between a heat pump and a sewage exchanger:

Variant I—heating of water used for cleaning stands, watering and preparing feed for pigs;
Variant II—heating water for irrigation in plant production;
Variant III—drying of cereals;
Variant IV—preparation of hot utility water for the breeder's residential building.

### 3.1. Variant I

Water on agri-breeding farms is used for two main purposes, i.e., drinking water and services water [30,31]. From an animal health point of view, it is beneficial for the drinking water to have a temperature just below the body temperature. This fact shows the need to heat it [32]. The classic approach to water preparation used so far by using electric or gas heaters is highly energy-consuming, and thus expensive. Reducing the costs of heating process water is possible by using a system employing waste heat from liquid manure. Figure 2 shows a diagram of the proposed water installation with a heat pump containing a sewage exchanger located in a closed underground slurry tank. The installation presented can be used to heat water for cleaning positions, watering and preparation of animal feed in pig farming.

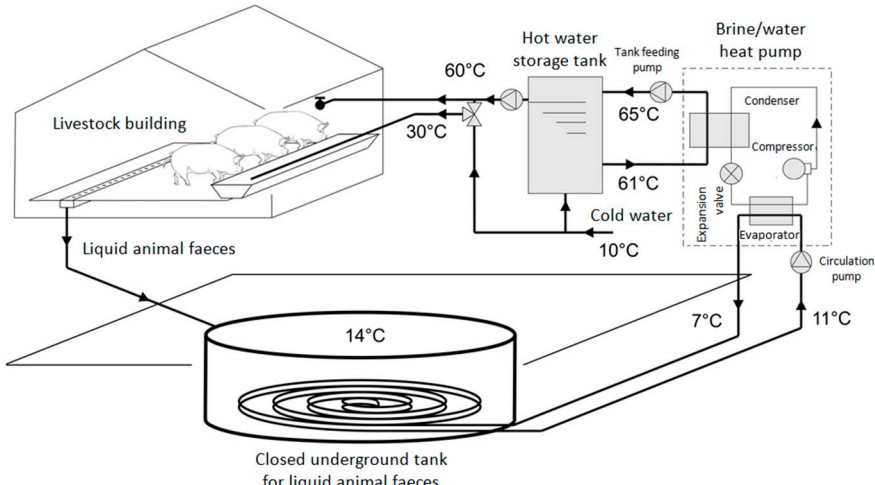

**Figure 2.** Diagram of the installation for heating water used for cleaning stands, watering and preparing animal feed with the use of a heat pump and a sewage exchanger.

The basic element of this installation is a closed slurry storage tank, with an average temperature of about 14 °C [29,33], which is the lower heat source. To transfer heat from the slurry to the evaporator, a low-solidifying liquid—brine or an aqueous glycol solution are applied. The heat carrier circuit is forced by the circulation pump. The upper source of the heat pump is the utility hot water storage tank exchanger. The heat supplied to the working medium initiates a change in the state of the refrigerant from liquid to gas. After passing through the evaporator, its temperature also rises. The refrigerant reaches the compressor, where its pressure and temperature increase. Before the condenser itself, it is a high temperature gas. In the heat exchanger, the heat is given off as a result of condensation, i.e., it changes its physical state again from gas to liquid. The last stage is expansion in the expansion valve, where the pressure is reduced to the initial level and the working medium in the liquid state reaches the evaporator [34]. The water heated in this way is then directed to drinking troughs, feed line installation and intake points in the livestock building.

### 3.2. Variant II

An optimal growth of the plant root system and preventing the emergence of conditions conducive to the development of fungal and bacterial diseases of the plant requires irrigation with water at a temperature close to summer ambient temperature, i.e., around 20 °C [35,36]. In traditional irrigation installations, water is heated using gas boilers and solid fuel. A heat pump installation can also be used for this purpose. The purpose of heating is to obtain water at an optimal temperature from the point of view of plant development. Ensuring adequate humidity requires dosing of water in a minimum of 10 L/m$^2$·day [37].

Modern irrigation systems allow precise determination of how much, at what intervals, at what pace, at what water temperature and how long specific batches of breeding are to be irrigated [38].

Figure 3 is a schematic diagram of a heat pump installation with a sewage exchanger for heating water for irrigation in plant production. The installation draws heat from the liquid manure using a sewage heat exchanger, which is located in a closed liquid manure tank below the surface. The heat pump, after receiving energy from the wastewater, transfers it to the hot water tank and then to the water sprinklers, which will spray the plants evenly.

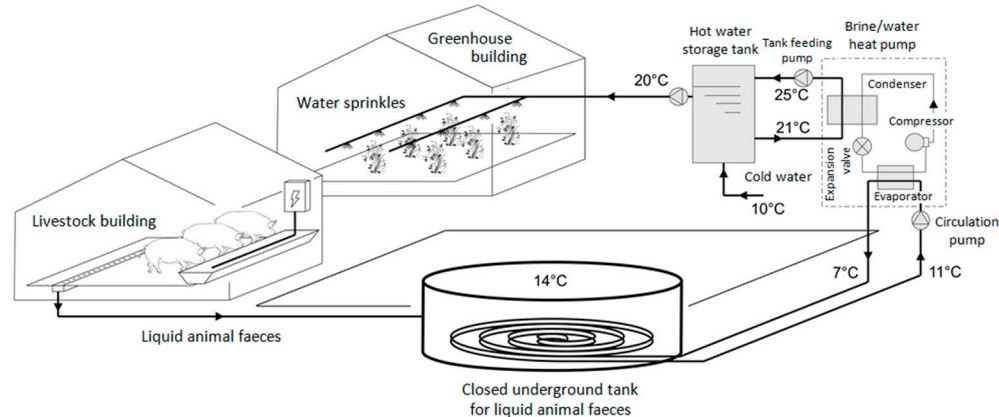

**Figure 3.** Diagram of the installation for heating water used for watering in plant production with the use of a heat pump and a sewage exchanger.

### 3.3. Variant III

In agricultural production, in many cases there is a need to dry cereal grains, feed, vegetable seeds and other agricultural products. Agricultural drying is one of the most energy-intensive processes of preserving agricultural products. The removal of water from a wide range of materials (fruit and vegetables, cereals) is dictated by many beneficial drying effects. First of all, there is a decrease in water activity and the release of chemical reactions, as well as the practical elimination of enzymatic reactions and the development of microorganisms, as a result of which the product is much more durable [39]. In addition, in most cases, the weight and volume of the materials dried are reduced, which facilitates and reduces packaging, transport and storage costs. Water contained in these raw materials immediately after harvest constitutes a significant percentage of their weight. Cereals, seeds and roots contain about 15–25% of water. Such a high water content and the presence of enzymes are the cause of rapid destruction of many substances after harvest, including biologically active substances. Therefore, the removal of water is necessary to stabilize active compounds. Drying consists in evaporating from the raw materials such a mass of water that its remaining content is on average from 7% to 14%. Only such humidity guarantees the cessation of the destructive activity of enzymes in dying cells. The medium used to dry a wide range of materials is hot air. By heating it, energy is obtained to evaporate moisture from the raw material. The air transfers thermal energy to the dried material, where it is used to evaporate water. It evaporates from the surface and interior of the material, and the resulting water vapor is transferred to the surrounding air. The drying medium is, therefore, used simultaneously to supply heat to the material dried and to remove the evaporated water. The drying medium should have such parameters and flow through the dryer in such an amount that it can absorb all the evaporated moisture, and in the case of convection drying—that it can transfer the required amount of heat to the raw material. In agriculture, flat-type drying devices, i.e., floor dryers, are mainly used. These solutions are characterized by low investments and low operating costs [40].

Figure 4 shows a schematic diagram of an installation with a heat pump and sewage exchanger used to heat air in the grain drying process. The installation draws heat from the liquid manure using a sewage heat exchanger, which is located in a closed liquid manure tank below the surface. It was assumed that the air is blown into the dryer building through a canal with a water air heater using an air supply fan. In this exchanger, air is heated with water connected to the heat pump's condenser circuit. Then the heated air is directed through the canal to the dryer through a metal plate with holes. The air flows through a layer of dried raw material (grain), and is then thrown out by an air duct through the exhaust fan. The heat pump works in a brine-water system.

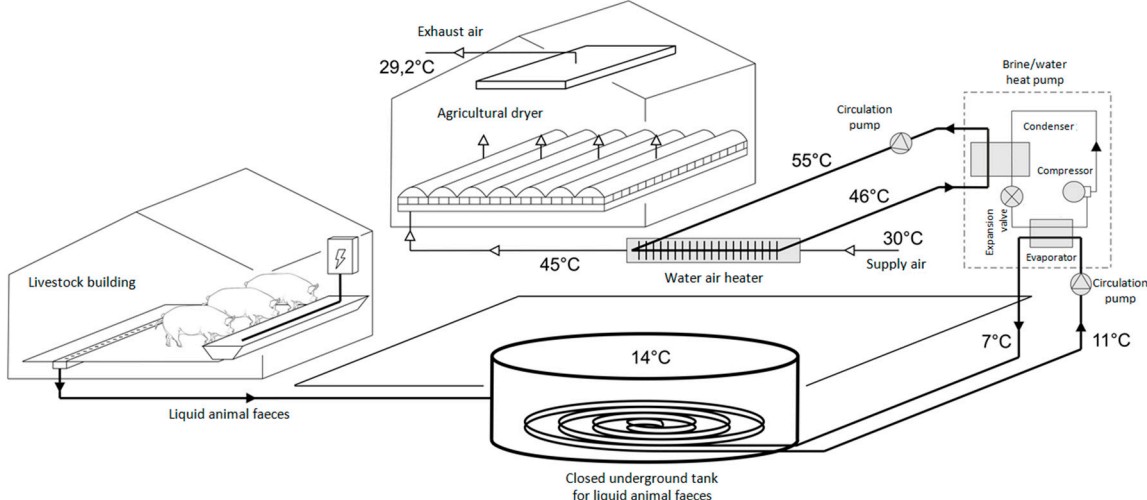

**Figure 4.** Diagram of cereal drying installation using a heat pump cooperating with a sewage exchanger.

*3.4. Variant IV*

The option proposes the use of waste heat from liquid manure to heat water in the farmer's apartment building. The preparation of hot water on farms for living purposes may constitute as much as 14.8% of the total energy consumption in these facilities [41].

The heat recovery system in this variant is shown in Figure 5. The heat is extracted by means of a brine circulating pump from wastewater at 14 °C from the underground slurry tank and transported to the heat pump evaporator. The compressor heat pump draws low-temperature energy and transfers it to the upper heat source—utility hot water installation. For the proper operation of the system, the use of a utility hot water tank is foreseen.

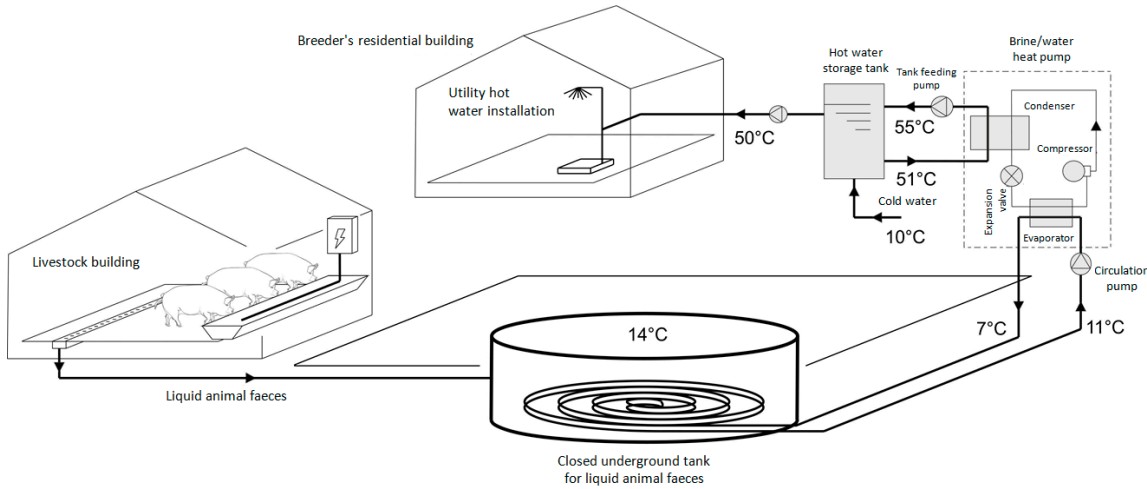

**Figure 5.** Installation diagram for preparing hot utility water in a farmer's apartment building with a heat pump and sewage exchanger.

## 4. Methods

The assessment of the energy and financial effects of using liquid manure heat-recovery systems first required determining the value of water consumption. For this purpose, based on the information contained in the literature [35,36,38,42–44] and using calculation Formulas (1) and (2), hydraulic parameters for individual variants can be determined.

Average hourly utility hot water demand $q_{srh}$ was calculated from the formula:

$$q_{srh} = \frac{q_{srd}}{t_e}$$ (1)

where:

$q_{srd}$—average daily demand for hot utility water, L/day,
$t_e$—the time of using hot water for specific purposes, h.

The maximum hourly utility hot water demand $q_{maxh}$ was calculated from the formula:

$$q_{maxh} = q_{srh} \times n_h$$ (2)

where $n_h$—hourly unevenness coefficient.

For such defined hydraulic parameters, the Formulas (3)–(5) can be used to determine the characteristic power necessary to heat the water to the desired temperature.

The average calculated power of the utility hot water system $Q_{srh}$ was calculated from the formula:

$$Q_{srh} = q_{srh} \times c_w \times (T_z - T_n)$$ (3)

where:

$c_w$—specific heat of water, J/kg·K,
$T_z$—heated water temperature, K,
$T_n$—municipal water temperature, K.

The maximum calculated power of the utility hot water system was calculated from the formula:

$$Q_{maxh} = q_{maxh} \times c_w \times (T_z - T_n)$$ (4)

The reduced calculated power of the utility hot water system was calculated from the Formula (5). It requires the determination of a reduction factor $\psi$ which determines the impact tray for domestic hot water preparation work. It allows the reduction of the maximum system power taking to be determined into account the heat accumulation in the tank.

$$Q_z = Q_{maxh} \times \psi$$ (5)

where: $\psi$—reduction coefficient, determined on the basis of [34].

In the case of variant III assuming the use of waste heat for drying agricultural products, the amount of moisture removed $W$ can be calculated from Formula (6).

$$W = G_p \times \frac{w_p - w_k}{100 - w_k}$$ (6)

where:

$G_p$—initial mass of moist material, kg,
$w_p$—initial moisture of material, %,
$w_k$—final moisture of material, %.

In turn, the Formula (7) can calculate the required air flow rate $L$.

$$L = l \times W$$ (7)

where *l*—specific dry air consumption as determined from Formula (8), kg/kg.

$$l = \frac{1}{x_w - x_z} \qquad (8)$$

where:

$x_w$—moisture content in the air after the drying process, g/kg,
$x_z$—moisture content in the air before the drying process, g/kg.

The proper heat consumption *q* can be calculated from Formula (9).

$$q = \frac{i_w - i_z}{x_w - x_z} \qquad (9)$$

where:

$i_w$—specific enthalpy of air after the drying process, kJ/kg,
$i_z$—specific air enthalpy before drying, kJ/kg.

The heat flow rate *Q* was calculated from Formula (10).

$$Q = W \times q \qquad (10)$$

where:

*W*—amount of moisture removed, kg/h,
*q*—proper heat consumption, kJ/kg.

The heat output of the heat pump $Q_{pc}$ for all variants of the heat recovery system can be calculated from Formula (11).

$$Q_{pc} = \frac{24}{24 - t_z} \times Q \qquad (11)$$

where:

*Q*—calculated demand for thermal power for preparing hot utility water, process water or drying agricultural products, kW,
$t_z$—tank utilization time (heat pump standstill), h.

The required heat exchanger length $L_w$ is given by Formula (12).

$$L_w = \frac{Q_d}{q_e} \qquad (12)$$

where:

$Q_d$—thermal power obtained from a low-temperature source, determined by the heat pump manufacturer, kW,
$q_e$—specific heat output taken from the ground, W/m$^2$.

In the case of systems using heat pumps and other unconventional sources, the period after which the economic benefits are noticed is relatively long, therefore, in financial analyses it is justified to use discount methods in calculations.

The assessment of the financial effectiveness of the application of individual variants of the heat recovery system from wastewater was carried out using the annual cost method. It is a discount method that allows for changes in the value of money over time to be taken into account. This is particularly important in the case of investments with long payback periods.

The capital return (extended reproduction) rate *r* was calculated from the formula [34]:

$$r = \frac{p \times (1+p)^N}{(1+p)^N - 1} \tag{13}$$

where:

*p*—discount rate, %,
*N*—calculated service life of the facility, years.

Formulas from 14 to 19 were used as proposed by Kusto [45,46].
The installment of fixed costs *r* + *r_ce* (the sum of the installment of the extended reproduction and the installment of the fixed operating costs) was calculated from the formula:

$$r + r_{ce} = 0.10719 \tag{14}$$

where:

*r_ce*—rate of fixed operating costs, %; 2% adopted as proposed by Kusto [46].

Annual fixed costs $K_{rst}$ were calculated according to Formula (15).

$$K_{rst} = K_{rr} + K_{est} = K_{inpc} \times r + K_{inpc} \times r_{ce} = (r + r_{ce}) \times K_{inpc} \tag{15}$$

where:

$K_{rr}$—total capital return costs, €/year,
$K_{est}$—the sum of fixed operating costs, €/year,
$K_{inpc}$—total capital expenditure on a heat pump installation; €.

In order to calculate the annual variable operating costs—the variable component of the annual costs of $K_{ezm}$ the Formula (16) was used.

$$K_{ezm} = \frac{Q_{pc} \times T_{ipc} \times c_{el} \times k_{mr}}{\varphi \times \eta_{sil}} \tag{16}$$

where:

$Q_{pc}$—heat pump installed power, kW,
$T_{ipc}$—heat pump installed power usage time, h/year,
$c_{el}$—electricity price, €/kWh,
$k_{mr}$—cost factor of moving materials for the heat pump, -; 1.02 adopted according to the Kusto proposals [46],
$\varphi$—average annual heating efficiency factor,
$\eta_{sil}$—efficiency of the electric motor driving the heat pump compressor, annual average value, %; 85% were adopted as Kusto proposed [46].

The annual costs of heat production $K_{rpc}$, as the sum of the constant component and the variable component were calculated from Formula (17).

$$K_{rpc} = K_{rst} + K_{ezm} \tag{17}$$

The amount of useful heat supplied by the heat pump during the year was calculated from Formula (18):

$$Q_{apc} = Q_{pc} \times T_{ipc} \tag{18}$$

The unit cost of heat generated $q_{pc}$ is the quotient of the annual costs and useful heat generated annually. This value can be determined based on the Formula (19).

$$q_{pc} = \frac{K_{rpc}}{Q_{apc}} \tag{19}$$

Using Formula (20), a reduction in the cost of obtaining heat $O$ for heating hot water was calculated for 1 kWh in relation to the cost of electricity.

$$O = c_{el} - q_{pc} \tag{20}$$

The financial savings obtained by the user as a result of operating the heat pump during the year were determined from Formula (21).

$$\Delta K = O - Q_{apc} \tag{21}$$

where:

$O$—reduction of costs of obtaining heat for heating hot water for 1 kWh in relation to the cost of electricity, €/kWh,
$Q_{apc}$—the amount of useful heat supplied per year by the heat pump, kWh/year.

A simple payback period for *SPBT* expenses incurred for building a system is determined by Formula (22).

$$SPBT = \frac{K_{inpc}}{\Delta K} \tag{22}$$

The environmental effect *EE* related to the investment outlays incurred should be determined from Formula (23), which describes the unit costs of *EE* pollution reduction.

$$EE = \frac{K_{inpc}}{Mp} \tag{23}$$

where $Mp$—weight of pollution resulting from the combustion of conventional fuel during the analysis period, kg.

## 5. Results and Discussion

Based on Formulas (1) to (5), the demand for thermal power for the preparation of utility hot water was calculated for variants I, II and IV. The results of the calculations are presented in Table 3.

**Table 3.** Summary of data and results of calculations of the demand for thermal power for the preparation of hot utility water in the analyzed agri-breeding farm.

|  | Variant I | Variant II | Variant IV |
|---|---|---|---|
| Number of inhabitants in the household (person) | 6 | 6 | 6 |
| Average daily demand for hot water $q_{śr\,d}$ (L/day) | 4000 | 5000 | 660 |
| Average hourly hot water demand $q_{śr\,h}$ (L/h) | 222.2 | 714.3 | 36.7 |
| Average power of the utility hot water system $Q_{śr\,h}$ (kW) | 12.8 | 8.3 | 17 |
| Maximum hourly utility hot water demand $q_{max\,h}$ (L/h) | 555.6 | *n/a* | 220.7 |
| Maximum power of the utility hot water system $Q_{max\,h}$ (kW) | 31.9 | *n/a* | 10.2 |
| Reduction factor $\psi$ (-) | 0.73 | 0.77 | 0.36 |
| Reduced system power for preparing utility hot water $Q_z$ (kW) | 26.1 | 7.1 | 4.1 |

*n/a*—not applicable.

Variant III, in which all waste energy is used for drying cereals, does not provide for the use of waste heat for the production of hot utility water, but only for hot process water. The obtained calculation results for option III constituting the basis for further analyzes are presented in Table 4.

**Table 4.** Comparison of data and results of calculations of the demand for thermal power for the preparation of hot technological water in the analyzed agri-breeding farm for the needs of drying agricultural products.

|  | Variant III |
|---|---|
| Initial weight of wet material $G_p$ (kg) | 7140 |
| Amount of moisture removed $W$ (kg/h) | 49.6 |
| Appropriate dry air consumption $l$ (kg/kg moisture drain) | 166.7 |
| Air flow rate $L$ (kg/h) | 8263.9 |
| Specific heat consumption $q$ (kJ/kg moisture drain) | 2533.3 |
| Heat flow rate $Q$ (kJ/h) | 125,611.1 |
| Calculated system power for preparing hot process water $Q_z$ (kW) | 34.9 |

Based on the results of calculations (Tables 3 and 4) and technical data available in the manufacturers' catalogs and publications, the heat output of the heat pumps was calculated, and then the devices operating in the brine/water system available on the market were selected for individual variants.

The sewage heat exchanger was designed in the form of GEO CALIX energy cages located in an underground closed liquid manure tank. Unit heat collection $q_e$ by PE 32 pipe was assumed at a safe level of 40 W/m. The results of calculations and selection of heat pumps for individual variants are summarized in Table 5.

**Table 5.** List of calculation results for the selection of heat pumps.

|  | Variant I | Variant II | Variant III | Variant IV |
|---|---|---|---|---|
| Calculated system power for utility hot water and process water $Q$ (kW) | 26.1 | 7.1 | 34.89 | 4.1 |
| Storage time $t_z$ (h) | 9 | 1 | n/a | 9 |
| Calculated heat output of heat pump $Q_{pc}$ (kW) | 41.7 | 8.2 | 34.9 | 6.6 |
| Selected heat pump | Dimplex SIH 40TE | STIEBEL ELTRON WPF 5 | STIEBEL ELTRON WPF 27 | Viessmann Vitocal 200-G |
| Heating capacity of the pump $Q_{pc}^{'}$ (kW) | 44.0 | 8.4 | 35.5 | 7.0 |
| Cooling capacity $Q_d$ (kW) | 29.3 | 7.2 | 26.2 | 5.0 |
| Electric power consumption $P$ (kW) | 14.7 | 1.2 | 9.3 | 2.0 |
| Coefficient of performance COP (-) | 2.8 | 7.5 | 3.7 | 3.3 |
| Length of sewage heat exchanger $L_w$ (m) | 732.5 | 180 | 655 | 125 |

Investment expenditures for individual system variants were determined using the calculations of companies dealing in the design and implementation of heat recovery systems, which are presented in Table 6.

**Table 6.** Investments for individual variants.

|  | Variant I | Variant II | Variant III | Variant IV |
|---|---|---|---|---|
| Heat pump (€) | 14,750 | 4040 | 9865 | 4950 |
| Heat exchanger (€) | 8780 | 2335 | 7660 | 1775 |
| Other elements of the installation: hot water tank, pipelines, fittings, automation system as well as labor and commissioning of the installation (€) | 5882.5 | 1593.75 | 4381.25 | 1681.25 |
| Total investments $K_{inpc}$ (€) | 29,412.5 | 7968.75 | 21,906.25 | 8406.25 |

For the prepared output data for the financial model described by Formulas (14) to (22), the values of its parameters were calculated. The results obtained are summarized in Table 7.

**Table 7.** Values of financial investment parameters.

|  | Variant I | Variant II | Variant III | Variant IV |
|---|---|---|---|---|
| Capital return installment $r$ | 0.0872 | 0.0872 | 0.0872 | 0.0872 |
| Installment of fixed costs $r + r_{ce}$ | 0.107 | 0.107 | 0.107 | 0.107 |
| Annual fixed costs $K_{rst}$ (€/year) | 3157.49 | 855.12 | 2351.21 | 903.25 |
| Annual variable operating costs $K_{ezm}$ (€/year) | 14,265.41 | 235.29 | 4658.03 | 1925.64 |
| Annual costs of heat generation $K_{rpc}$ (€/year) | 17,422.90 | 1088.07 | 7009.25 | 2828.89 |
| Amount of useful heat $Q_{apc}$ (kWh/year) | 240,900 | 10,642.8 | 103,944 | 38,325 |
| Unit cost of heat generated $q_{pc}$ (€/kWh) | 0.07 | 0.10 | 0.07 | 0.07 |
| Reduction in the cost of obtaining heat $O$ (€/year) for 1 kWh | 0.07 | 0.04 | 0.07 | 0.07 |
| Financial savings $\Delta K$ (€/year) | 15,863.04 | 380.14 | 7353.02 | 2466.60 |
| *SPBT* (years) | 1.9 | 21 | 3 | 3.4 |

When analyzing the results presented in Table 6 for Variants I to IV, it can be stated that Variant I is characterized by the highest investments where the heat pump is used to heat water for drinking and preparing animal feed in pig farms. The expenditure is €29,412.5. In turn, Variant II of the system has the lowest investment costs of €7968.75. Therefore, the difference in capital expenditure incurred is very significant and amounts to nearly 270%. The differences in investments for individual variants are correlated with the required heat output and the size of the installation.

It is obvious that the sheer value of investments should not be a factor that would determine the investment decision. Operating costs are also important, as are the finally acceptable payback period. As shown in Table 7, Variants I and III have the highest operating costs. For Variant I it is €14,265.41 and for Variant III €4658.03. The high operating costs are due to the high power output of the heat pump compressors, and thus to the high consumption of electricity.

When analyzing the payback period, it can be observed that for Variants I, III and IV it is relatively short and amounts to less than 4 years. Interestingly, in the case of Variant I, which was characterized by the highest investment and operating costs, the payback period is even shorter and amounts to 1.9 years. Thus, this is an extremely favorable situation and clearly speaks for the implementation of this option by the investor. In turn, Variant II, for which investments and operating costs reached the lowest values, is the least favorable option from the investor's point of view, the payback period for this option is 21 years and exceeds the assumed durability of the heat recovery system. Therefore, it is a variant that should not be recommended for implementation by the investor. Figures 6 and 7 show the return on investment over time.

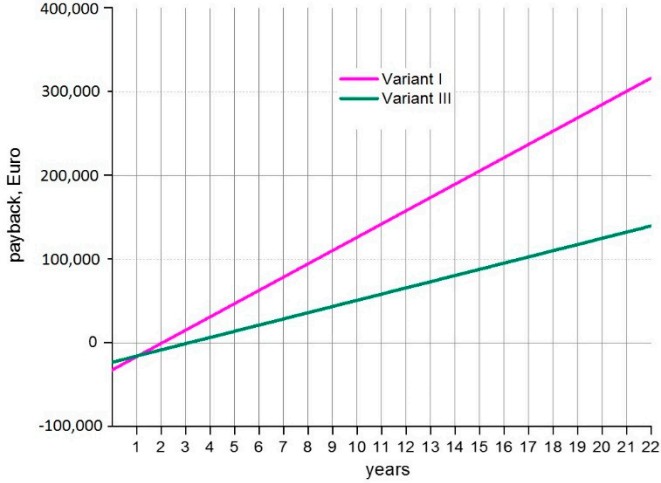

**Figure 6.** Reimbursement of investment costs of heat recovery systems in agri-breeding farm for variants I, III.

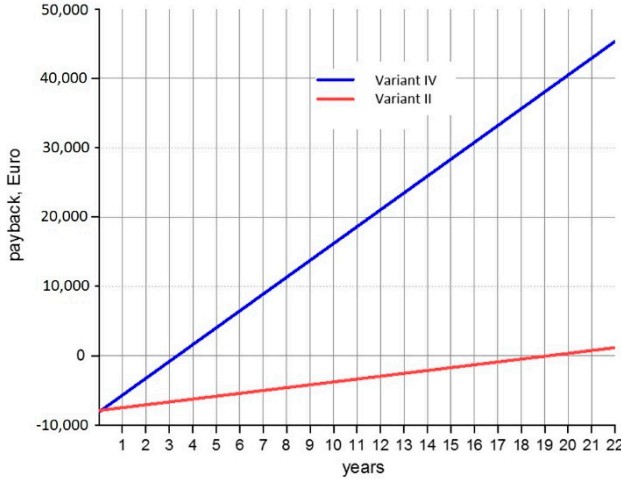

**Figure 7.** Reimbursement of investment costs of heat recovery systems on an agro-animal farm for variants II, IV.

Hot water can be prepared using electricity obtained from unconventional or traditional sources using fossil fuels [47]. The paper analyzes the possibility of reducing pollutant emissions as a result of the use of waste heat in relation to conventional heat sources using coal and gas as fuel. The energy yield $E_g$ was determined as the difference between the energy needed for the operation of the heat pump $E_e$ and the amount of useful heat energy supplied by the pump to heat the $Q_{apc}$ water. Table 8 summarizes the energy yield values $E_g$.

**Table 8.** Energy yield.

|  | **Variant I** | **Variant II** | **Variant III** | **Variant IV** |
|---|---|---|---|---|
| Annual electricity consumption by the heat pump (kWh/year) | 96,645 | 7889 | 61,143 | 13,149 |
| Energy yield (kWh/year) | 144,255 | 2754 | 42,801 | 25,176 |

Considering that the burning of 1 kg of coal provides 2.460 kWh of energy [48], and 1 kg of natural gas 13.5 kWh [49], calculations were made of the amount of these fuels that would be able to meet the heat demand in individual system variants. The results of the calculations are summarized in Table 9.

**Table 9.** The amount of coal and natural gas necessary to produce heat energy replaced by waste.

|  | **Variant I** | **Variant II** | **Variant III** | **Variant IV** |
|---|---|---|---|---|
| Coal (kg) | 58,640 | 1120 | 17,399 | 10,234 |
| Natural gas (kg) | 10,686 | 204 | 3170 | 1865 |

It is known that the burning of 1000 kg of hard coal in order to generate thermal energy causes, on average, the emission of: 9.6 kg $SO_x$, 3.2 kg $NO_x$, 10 kg CO, 2130 kg $CO_2$, 10 kg PM (particulate matter) suspended in exhaust gases and 0.003 kg benzo(a)pyrene the reduction of emitted pollutants resulting from the use of waste heat was determined [50,51]. Similar calculations were made assuming that thermal energy for the preparation of hot water, process water and for drying agricultural products will be produced as a result of burning natural gas. It was assumed, according to generally available data, that as a result of burning 1 m$^3$ of gas, flue gas is emitted containing: $8 \times 10^{-6}$ kg $SO_x$, $1.65 \times 10^{-3}$ kg $NO_x$, $0.3 \times 10^{-3}$ kg CO, 2 kg $CO_2$. The obtained calculation results are presented in Figures 8 and 9.

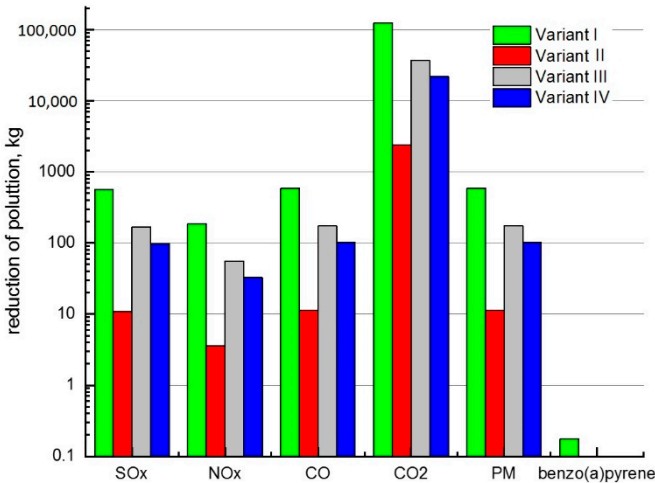

**Figure 8.** Pollution reduction associated with the use of waste heat on an agricultural and livestock farm as an alternative to heat production as a result of coal burning.

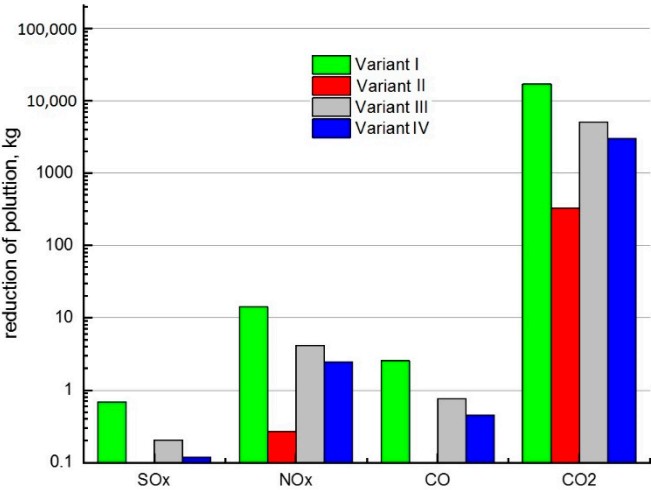

**Figure 9.** Pollution reduction associated with the use of waste heat on an agri-breeding farm as an alternative to heat generation as a result of gas burning.

The highest pollution reduction was obtained for variant I and amounted to 124,903 kg $CO_2$ in the case of coal replacement with waste energy, and 17,098 kg $CO_2$ in the case of natural gas. In the case of the other variants, the reduction of impurities was significantly lower. Variant I was also characterized by the largest reductions in other impurities. In the case of replacement of hard coal as fuel by waste heat, an additional reduction of 563 kg $SO_x$, 187 kg $NO_x$, 586 kg CO, 586 kg dust and 0.18 kg benzo(a)pyrene was achieved. In the case of natural gas, the use of waste heat has reduced emissions by an additional 0.68 kg $SO_x$, 14.11 kg $NO_x$, 2.56 kg CO. As the results of the research show, these reductions are significant.

In order to analyze the environmental effect related to the incurred financial expenses, the unit costs of reduction of *EE* pollutants as a result of using waste heat as a source of heat energy were calculated based on Formula (23). The results of the calculations are presented in Figures 10 and 11.

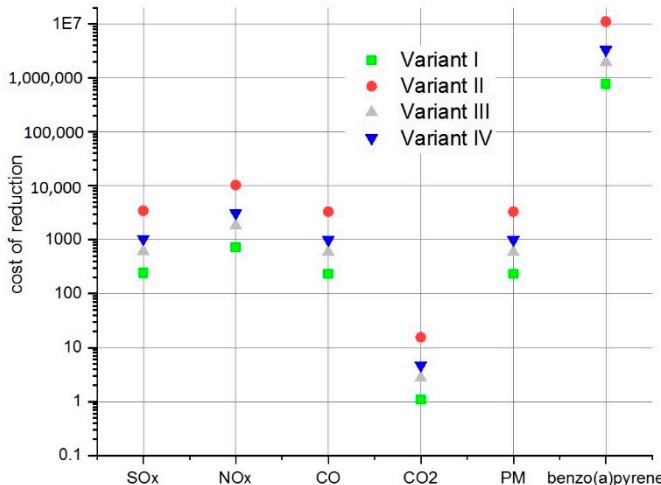

**Figure 10.** Cost of reduction of 1 kg of pollutants in relation to coal as a fuel for heat generation.

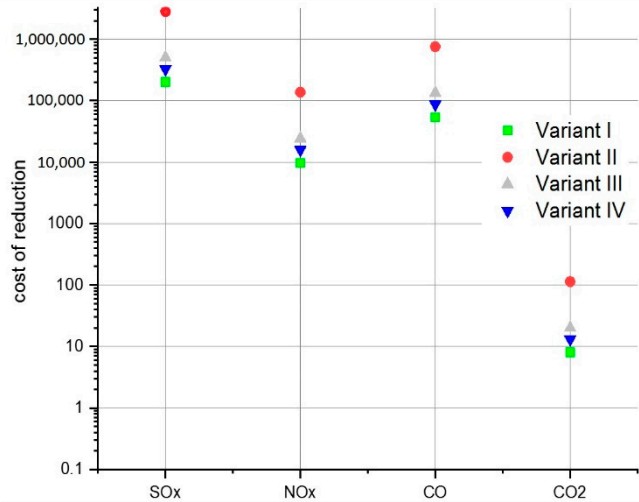

**Figure 11.** Cost of reduction of 1 kg of pollutants in relation to gas as a fuel for heat generation.

Variant I was characterized by the lowest unit costs of reduction of all *EE* pollutants, both in terms of fuel in the form of hard coal and natural gas. Therefore, it is the most favorable option in terms of the payback period of simple payback time (*SPBT*) expenses as well as in terms of the most favorable environmental impact. This option as the most financially and environmentally effective should be recommended to the investor.

## 6. Conclusions

To sum up, it can be stated that the use of thermal energy from liquid manure as a waste heat source in agri-breeding farms is technically possible, financially profitable and also beneficial for the natural environment. The results of calculations of the financial efficiency of using the heat recovery system presented in the article justify the purposefulness of its use in order to reduce the amount of energy consumed from conventional sources.

The legitimacy of the use of individual variants is associated with the farm's operating characteristics and the method of managing the recovered heat energy. The payback time for investments for the slurry heat-recovery installation depends on the system variant under consideration and adopts very wide limits from around 2 to 21 years. Such large differences in the value of the *SPBT* parameter should prompt investors to conduct a reliable financial analysis of the investment. The apparent advantages in the form of low investment outlays and low operating costs, as proved

in the studies, do not correspond to the most favorable investment option and the largest financial savings as well as the shortest payback period.

The research also determined the approximate ecological effect in the form of reduction of pollution resulting from the combustion of conventional fuels in order to generate thermal energy for the preparation of hot utility water and process water by replacing them with waste energy recovered from slurry. Obtaining thermal energy from this source has a number of advantages, including reducing the dynamics of matter decomposition processes and slowing down the formation of gases, including greenhouse gases, while being characterized by high temperature stability and significant quantitative potential.

The results of pollution reduction due to the use of waste heat sources are very promising, especially if waste heat replaces conventional methods of preparing hot water by energy obtained from burning fossil fuels. The environmental effect will, of course, be significantly lower when using unconventional energy sources in the form of solar or wind energy. Nevertheless, energy saving and the widest possible use of waste energy, including thermal energy, is a very important element of sustainable energy management and reduction of the negative environmental impact of the economy.

**Author Contributions:** Conceptualization, D.S. and D.C.; methodology, D.S. and D.C.; software, K.P.; validation, D.S. and K.P.; formal analysis, D.S. and D.C.; investigation, K.P.; resources, K.P.; data curation, D.C.; writing—original draft preparation, D.S. and K.P.; writing—review and editing, D.S. and K.P.; visualization, D.S. and K.P.; supervision, D.S.; project administration, D.S. and K.P.; funding acquisition, D.S. All authors have read and agreed to the published version of the manuscript.

**Funding:** This research received no external funding.

**Conflicts of Interest:** The authors declare no conflict of interest.

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
