# Peer review of "An Analysis of Waste Heat Recovery from Wastewater on Livestock and Agriculture Farms"

_resources, doi:10.3390/resources9010003_

Round 1
Reviewer 1 Report
Well done paper dealing with usage of waste heat recovery system on farms and farming.
Introduction part is quite well done however deeper literature overview should be done. It continue only general information that agriculture is treat to the nature or that there is a trend of high interest in usage of waste energy sources of some general fact about sustainable development. There should be mentioned more papers and studies which were namely dealing with the analysis of waste heat recovery or they used similar methodology of energy efficiency test or financial efficiency test…..what was the conclusion in their research? How many variants they assessed? Is it possible to choose method which leads to the most promising evaluations and conclusion? There is an overview through the world wide (Belgium, England, Spain, Mexico…), but some papers dealing with the situation in Poland (except author´s papers) should be implemented too (if it is possible). Are there any others research groups in Poland dealing with the energy and financial efficiency tests?
Authors should extent the Introduction part.
Test object and input data: Input data are clearly defined and described in connection to the next heading -Variants. Every chosen variant is appropriately supported with scheme/diagram of the installation.
Methods proceeded from water consumption for energy and financial effect assessment is presented via many available formulas which were suitably chosen for calculation.
Anyway in this part some inaccuracies were noticed. To avoid to decreasing of the scientific sound of the paper I recommend correcting it in whole text.
Line 147 - 153: check the alignment of the paragraph
Line 219: correct the unit for average daily demand [L/d] …and replace letter “l” for “L” as it is used in the rest of the paper
Line 219, 220, 230, 231, 241-243…, 271…… and the rest of the paper: use square bracket for the units [h], [K], [%],[kg/kg]…
Line 227 …. fill symbol “Qsrh” (average power of the utility hot water system) to the sentence
Line235 - formula (5) …..symbol “=” is missing in the formula - fill it please,
Line 239: correct the symbol “W” to italic type - “W” to unify it with the rest of the paper,
Line 256: fill symbol “Q” (heat flow rate) to the sentence
Line 267: fill and correct lower symbol “w” symbol to “L” (length of heat exchanger) to the sentence
Line 283: r + re write in italic style
Line 289, 290, 291, 293: symbols “K” write in italic style (“K”)
Line 305: The amount of useful heat …….supplied by the heat pump during - insert symbol Qapc where the dots are
Line 317 Qapc - write in italic style
Line 320: The environmental effect ……. related to - insert symbol EE where the dots are
Line 317, 322: incorrect paragraph indent - unify it with the rest of the text
Line 309 and 310: check the alignment of the sentence
Table 7: units - kg …….the square brackets are missing - fill it please
Line 400 and 401: correct the notation of contain of pollutants in the flue gas (use upper index, use “.” or “x” for multiply symbol (instead of big black dot in the middle of the line)… see also line 153, 122,
Line 425: The abbreviation SPBT should be explained where is first mentioned in the text - so it should be here, for the other readers (it is used for the first time here; so:”…..terms of payback period of simple payback time (SPBT) expenses as well as ……..”
I guess that it should be simple payback time?
Results and Conclusion:
Parts “Results” are presented clearly with the “Conclusion” supported consequently with the high originality degree. It was proven that the using of thermal energy from liquid manure as a waste heat source on farming is technically possible, financially profitable and also it reduce the air pollution.
I recommend the paper for publishing in MDPI journal Resources after minor revision according to comments written above. The importance of this work cannot be negligible from the savings of energy and reduction in pollution by using of waste heat recovery system´s point of view.
Author Response
Reviewer #1
The authors thank the Reviewer #1 for taking the time to evaluate the manuscript and we appreciate his remarks. We took in to account all comments but here we answered for the most important.
Comment 1: Introduction part is quite well done however deeper literature overview should be done. It continue only general information that agriculture is treat to the nature or that there is a trend of high interest in usage of waste energy sources of some general fact about sustainable development. There should be mentioned more papers and studies which were namely dealing with the analysis of waste heat recovery or they used similar methodology of energy efficiency test or financial efficiency test…..what was the conclusion in their research? How many variants they assessed? Is it possible to choose method which leads to the most promising evaluations and conclusion? There is an overview through the world wide (Belgium, England, Spain, Mexico…), but some papers dealing with the situation in Poland (except author´s papers) should be implemented too (if it is possible). Are there any others research groups in Poland dealing with the energy and financial efficiency tests?
Response: We are grateful for this comment. We have extended the introduction part and we add Table 1 in the line 88. This Table includes research of other concerning waste heat recovery.
Comment 2: Test object and input data: Input data are clearly defined and described in connection to the next heading -Variants. Every chosen variant is appropriately supported with scheme/diagram of the installation.
Response: We are grateful for this comment.
Comment 3: Methods proceeded from water consumption for energy and financial effect assessment is presented via many available formulas which were suitably chosen for calculation.
Response: We are grateful for this comment.
Comment 4: Line 147 - 153: check the alignment of the paragraph; Line 309 and 310: check the alignment of the sentence
Response: We are grateful for this comment. We have corrected it.
Comment 5: Line 219: correct the unit for average daily demand [L/d] …and replace letter “l” for “L” as it is used in the rest of the paper
Response: We are grateful for this comment. We have replaced l for L in whole article.
Comment 6: Line 219, 220, 230, 231, 241-243…, 271…… and the rest of the paper: use square bracket for the units [h], [K], [%],[kg/kg]… ; Table 7: units - kg …….the square brackets are missing - fill it please
Response: Thank you for your comments. We have corrected the units.
Comment 7: Line 227 …. fill symbol “Qsrh” (average power of the utility hot water system) to the sentence
Response: Thank you for your comments. We add this information in the line 221. “
The average calculated power of the utility hot water system Qsrh was calculated from the formula:“
Comment 8: Line235 - formula (5) …..symbol “=” is missing in the formula - fill it please,
Response: Thank you for your comments. We add missing symbol.
Comment 9: Line 239: correct the symbol “W” to italic type - “W” to unify it with the rest of the paper,
Response: Thank you for your comments. We have corrected it.
Comment 10: Line 256: fill symbol “Q” (heat flow rate) to the sentence
Response: Thank you for your comments. We add this information in the line 253. “The heat flow rate Q was calculated from the relation (10).”
Comment 10: Line 267: fill and correct lower symbol “w” symbol to “L” (length of heat exchanger) to the sentence
Response: We are grateful for this comment. We have corrected the symbol. Actual sentence:
“The required heat exchanger length Lw is given by formula (12).”
Comment 11: Line 283: r + re write in italic style; Line 289, 290, 291, 293: symbols “K” write in italic style (“K”); Line 317 Qapc - write in italic style
Response: We are grateful for this comment. All mistakes was corrected.
Comment 12: Line 305: The amount of useful heat …….supplied by the heat pump during - insert symbol Qapcwhere the dots are; Line 320: The environmental effect ……. related to - insert symbol EE where the dots are; Line 317, 322: incorrect paragraph indent - unify it with the rest of the text; Line 400 and 401: correct the notation of contain of pollutants in the flue gas (use upper index, use “.” or “x” for multiply symbol (instead of big black dot in the middle of the line)… see also line 153, 122,
Response: We are grateful for this comment. We have corrected it.
Comment 13: Line 425: The abbreviation SPBT should be explained where is first mentioned in the text - so it should be here, for the other readers (it is used for the first time here; so:”…..terms of payback period of simple payback time (SPBT) expenses as well as ……..”
I guess that it should be simple payback time?
Response: We are grateful for this comment. We have add to the sentence simple payback time (SPBT) in the line 418
Comment 14: Results and Conclusion: Parts “Results” are presented clearly with the “Conclusion” supported consequently with the high originality degree. It was proven that the using of thermal energy from liquid manure as a waste heat source on farming is technically possible, financially profitable and also it reduce the air pollution.
Response: We are grateful for this comment.
Comment 15; I recommend the paper for publishing in MDPI journal Resources after minor revision according to comments written above. The importance of this work cannot be negligible from the savings of energy and reduction in pollution by using of waste heat recovery system´s point of view.
Response: Thank you very much for this valuable comment.
Reviewer 2 Report
The paper presents a technical and economical study regarding the use of waste heat in the Agriculture sector, for technological processes within an agri-breeding farm, respectively the methodology for developing the recovery processes envolving heat pump's technique with intermediary heat exchangers. This topic is timely and in the scope of the journal Resources.
The paper is a technical study, rather than a scientific one. Anyway, the study validates important hypotheses, such as - these kinds of applications are technically posssible, financially efficient and environmentally friendly. The paper validates once more that the ˮAgriculture is one of the sectors of economy in which it is possible to conduct more rational energy economyˮ.
In my personal opinion, the study is missing it's final part - a methodology based on multicriteria evaluation, with the scope to select the best solution from all suggested variants related to the analysed study case, respectively which one will support the decision making on different technical solutions, in order to increase the applicability of the study.
Please state clearly the place of the present paper in the scientific context, the targeted objectives and the novelty elements of the paper.
While the idea and construction of the manuscript are relevant, timely and interesting, from the current presentation it is not entirely clear which are the originality aspects of this paper. Thus, it is very hard to assess the merits of this manuscript, so I have to recommend the authors to emphasise more the elements of originality within the Introduction section, especially in relation with earlier papers cited in the text.
Here is a list of some suggestions which could be part of an improved manuscript:
Line 25-26 - Please revise it, in terms of expression. Line 36, Line 40 - Because the Introduction part represents the background that sustains the importance of this research paper and its objectives, it is important to clearify the role of the paper at first. In this regard, please explain what do you mean by ˮdifficult challengesˮ, and ˮthreatˮ, and further, what are the gaps that your research paper is trying to fill in the scientific context. It is not a practice to send the readers to find this information in other publications, in order to understand the specific context of the present research. Also, please specify the role of these ˮchallengesˮ in establishing the objectives of the paper. Test object and input data – This part will be more clear if the input data were presented in a centralised way, for example in a table, in which they are separated on types: constructive, functional, characteristics, parameters (for example). Also, please introduce in this section all the hypotheses that your research relies on (for ex. the heat exchange between the tank and the ground / environment is neglected). Correct l/d, or L/d (for example, Line 96-97, 153, etc.). For numbers, choose the type of separation for the thousand's digits (none, space, comma – acoording to the template), and use it unitarily throughout the text.Line 104 – Change option with variant. Line 147-153 – Please correct the paragraph's alignment. Explain the acronym DHW (domestic hot water) after its first aparition in the text. Line 106 and Line 132, the same assumption is atributed to two different references [19] and [23]. Please clearify this - [19], [23] or both. Line 217, 223 - How were te, nh taken into account (for example, from internal standards, experimental data, consumer profiles, etc.) ? Line 229 – The unit measure is missing. Line 236, Line 330 - Table 1 – φ is reduction coefficient (please explain in few words what does it represent and how is it determined). In Line 300 - φ is average annual heating efficiency factor. In Table 3, φ is degree of efficiency. Line 342-343 – Please specify on what basis was this assumption made (calculation, tests, manufacturer technical specifications, other researches, etc.) In Table 3, what are the operating efficiencies of the heat pumps taken into account (regarding the fact that different pumps have been chosen as thermal power and as a manufacturer)? How do these choices affect the secured results?
Please review all of the text and the paper too, in order to be in accordance with the journal template.

Author Response
Reviewer #2
The authors thank the Reviewer #2 for taking the time to evaluate the manuscript and we appreciate his remarks. We took in to account all comments but here we answer for the most important. We took in to account all comments but here we answered for the most important.
Comment 1: In my personal opinion, the study is missing it's final part - a methodology based on multicriteria evaluation, with the scope to select the best solution from all suggested variants related to the analysed study case, respectively which one will support the decision making on different technical solutions, in order to increase the applicability of the study.
Response: We are grateful for this comments. You have right we want to do this in the next manuscript. We will prepare survey and we want to choose appropriate experts to conduct multi-criteria analysis.
Comment 2: While the idea and construction of the manuscript are relevant, timely and interesting, from the current presentation it is not entirely clear which are the originality aspects of this paper. Thus, it is very hard to assess the merits of this manuscript, so I have to recommend the authors to emphasise more the elements of originality within the Introduction section, especially in relation with earlier papers cited in the text.
Response Thank you for this comments. We add Table 1 which include enlarger review in the research filed concerning waste heat recovery in agriculture.
Comment 3: Please state clearly the place of the present paper in the scientific context, the targeted objectives and the novelty elements of the paper.
Response: Thank you for this comments. We add more information and describes in the line 91-102
Comment 4. In Table 3, what are the operating efficiencies of the heat pumps taken into account
(regarding the fact that different pumps have been chosen as thermal power and as a manufacturer)?
Response: We are grateful for this comments. We add information in the lines 226-229. It was selected depending on the structure of the installation based on Kusto's proposals [45, 46]
Comment 5: Test object and input data – This part will be more clear if the input data were presented in a centralised way, for example in a table, in which they are separated on types: constructive, functional, characteristics, parameters (for example).
Response: We are grateful for this comments. We add Table 2 in which we present all input data
All other suggestion/ comments were taking in to account. We hope that our corrections and answers will be satisfactory.
Reviewer 3 Report
- The sentence which begins in Line 42 should be rephrased because it is hardly understandable. What means word acquis (Line 44)?
- Line 58 – citation should be written as [10-12].
- Line 67 – what it means sentence: “An important environmental aspect is worth noting. ”?
- Line 97 – it should stand l/d not only l d .
- Line 175 - “Cereals, seeds and roots contain about 15-25%. ” - the sentence should be “Cereals, seeds and roots contain about 15-25% of water.”
- Line 189 – instead of “investments ” should be written “low investments ”.
- Line 194 – instead of “it is heated ” it should be written “air is heated ”.
- In equation 5 is missing equal mark (=).
- Symbols in equation 9 should be italic.
- Symbols and markings throughout the paper text should be unified. For example – for q in Line 251 is stated “actual heat consumption”, but in Line 259 is stated “proper heat consumption ”. Please unify all the markings, symbols and its explanations throughout the paper text.
- Line 286 – instead of “proposed ” should be written “proposed in”.
- Line 296 - “h year ” or “h/year ”?
- Equations from 14 to 19 were used as proposed in [36]. It will be much better to state this fact before equation 14, and then avoid referencing [36] for each equation. As presented it is quite annoying to see reference [36] continuously for several times.
- Table 3 – data from the first row in this table should be equal to Qz from table 1 and table 2, but is not equal for the Variant IV. Is this typing mistake or have an additional explanation (which is not included in a paper text)? How this illogicality influences all obtained results?
- Table 3 – the Authors should explain “Degree of efficiency”, what it represents and how is it obtained. How it can be equal to 33 (Variant IV)?
- Table 5 – for O – it is not 1 KW, it is 1 kWh. Again, carefully check and correct all the markings throughout the paper text.
- Figure 6 – in the paper text is missing call on this figure.
- There are many wrong or impossible elements presented in Figure 5 and Figure 6. Figure 5 – Variant I – payback period starts after the first year – why not at the beginning of presented system operation? For this variant I – between second and third year of system operation will be made a profit of more than 300000 Euro? This is impossible. If this fact can have an explanation, the Authors presented the best possible way to become a millionaire.
Figure 6 – for Variant II, according to a figure – initial investment is zero, and the payback period is, as stated in a paper text – 21 years? Quite unusual... As in Figure 5, here occurs some enormous mistake – please, prepare these two figures carefully.
- Line 388 - “Considering that the burning of 1 kg of coal provides 2.460 kWh of energy, and 1 kg of natural gas 13.5 kWh... ” needs a reference or additional data, because this fact can be approximately true, but surely not for any coal or natural gas of any composition.
- Lines from 394 to 401 – presented data requires references which will confirm them or presentation of calculation procedure which will show how these data are obtained (both for coal and natural gas). According to my experience, without references which will confirm these data – this emission values may and may not be correct (especially when the specifications of both fuels is missing). Also, if the references for those values existed, the numbers should be properly written (exponents).
- Figure 7 and Figure 8 – again, many problems occur in those two figures. Firstly, if the presented data (comment above) are correct, it should be highlighted that presented reduction in emissions is in a whole year period. In the legend of both figures – why VAriant II, not as for others Variant II? In Figure 7 is presented PM (Particulate Matter), but in the paper text the Authors talk about dust. CO2 – 2 in the index (x axis). Figure 8 title – it should be specified that it is natural gas, as written only gas – it can be any else flammable gas.
- Line 413 - “ with respect to coal as fuel ” should be removed.
- Line 415 – instead of “very significant ” should be written only “significant ”.
- Figure 9 – benzo(a)pyrene cost of reduction for Variant II and IV is not visible (or partially visible). The same is in Figure 10 for SOx and Variant II. Again, x-axis, CO2-2 in the index.
- Reference [37] is not mentioned anywhere in the paper text.
- English throughout the paper text is readable and understandable, but it can and should be improved.
I encourage the Authors to perform detail and complete corrections of this paper. According to my opinion, it is highly valuable paper, but it must be notably improved, especially the parts which are related to payback period and emissions.
Author Response
Reviewer #3
The authors thank the Reviewer #3 for taking the time to evaluate the manuscript and we appreciate his remarks. We took in to account all comments but here we answer for the most important. We took in to account all comments but here we answered for the most important.
Comment 1: - Line 67 – what it means sentence: “An important environmental aspect is worth noting. ”?
Response: We are grateful for this comments. It was our mistake it should be noticing. It was corrected.
Comment 2: - Equations from 14 to 19 were used as proposed in [36]. It will be much better to state this fact before equation 14, and then avoid referencing [36] for each equation. As presented it is quite annoying to see reference [36] continuously for several times.
Response: Thank you for this comments. We add tis references in the beginning.
Comment 3: - Table 3 – data from the first row in this table should be equal to Qz from table 1 and table 2, but is not equal for the Variant IV. Is this typing mistake or have an additional explanation (which is not included in a paper text)? How this illogicality influences all obtained results?
Response: Thank you for this comment. You have right. That was our mistake. We have corrected it.
Comment 4: - Table 3 – the Authors should explain “Degree of efficiency”, what it represents and how is it obtained. How it can be equal to 33 (Variant IV)?
Response: Thank you for this comment. We changed the symbol it is COP (coefficient of performance). When it comes to value 33. It was mistake. It should be 3.3
Comment 5: - There are many wrong or impossible elements presented in Figure 5 and Figure 6. Figure 5 – Variant I – payback period starts after the first year – why not at the beginning of presented system operation? For this variant I – between second and third year of system operation will be made a profit of more than 300000 Euro? This is impossible. If this fact can have an explanation, the Authors presented the best possible way to become a millionaire.
Figure 6 – for Variant II, according to a figure – initial investment is zero, and the payback period is, as stated in a paper text – 21 years? Quite unusual... As in Figure 5, here occurs some enormous mistake – please, prepare these two figures carefully.
- Figure 7 and Figure 8 – again, many problems occur in those two figures. Firstly, if the presented data (comment above) are correct, it should be highlighted that presented reduction in emissions is in a whole year period. In the legend of both figures – why VAriant II, not as for others Variant II? In Figure 7 is presented PM (Particulate Matter), but in the paper text the Authors talk about dust. CO2 – 2 in the index (x axis). Figure 8 title – it should be specified that it is natural gas, as written only gas – it can be any else flammable gas.
Response: Thank you for this very valuable comments. The figures was corrected.
Comment 6: - Line 388 - “Considering that the burning of 1 kg of coal provides 2.460 kWh of energy, and 1 kg of natural gas 13.5 kWh... ” needs a reference or additional data, because this fact can be approximately true, but surely not for any coal or natural gas of any composition.
Response: We are grateful for this comments. We add references [4] and [49]
Comment 7: - Lines from 394 to 401 – presented data requires references which will confirm them or presentation of calculation procedure which will show how these data are obtained (both for coal and natural gas). According to my experience, without references which will confirm these data – this emission values may and may not be correct (especially when the specifications of both fuels is missing). Also, if the references for those values existed, the numbers should be properly written (exponents).
Response. Thank you for this very valuable comments. The add literature [50 and 51]
Comment 8: I encourage the Authors to perform detail and complete corrections of this paper. According to my opinion, it is highly valuable paper, but it must be notably improved, especially the parts which are related to payback period and emissions.
Response: We are grateful for this comments. Thank you for your time and valuable comments. We hope that our corrections and answers will be satisfactory.
Round 2
Reviewer 2 Report
The Reviewer appreciates that his remarks were taken into account and most of the comments were well received by the Authors.
The Authors took into account almost all comments and answer for the most important ones regarding the research background and methodology. The authors have improved the Introduction part, emphasized more the role of the present paper in the scientific context and structured the input data. They also made the corrections regarding the notations and units of measure related to the used physical quantities.
In this regard, I recommend this paper for the publication in Resources Journal, after minor revision regarding text editing.
I also wish success to the authors in their endeavors and I would like to encourage them to continue their study with concretization in a future manuscript.
Kind regards,
Reviewer

Reviewer 3 Report
The Authors performed all the required corrections. Now I consider this paper as completely correct, interesting to a wider audience and innovative. I don't understand how some obvious mistakes can occur in the first paper version, but ok - all of them are corrected in this revised version. I hope that the Authors will continue its research about such (or similar) systems. Keep up a good work and avoid obvious mistakes. At the end, I can state - good engineering and scientific job!